# Oral health treatment habits of people with schizophrenia in France: A retrospective cohort study

Frédéric Denis[1,2,3]*, Karine Goueslard[4,5], Francesca Siu-Paredes[6,7], Gilles Amador[2], Emmanuel Rusch[1], Valérie Bertaud[8,9], Catherine Quantin[4,5,10,11]

1 Faculté de Médecine, EA 75–05 Education, Ethique, Santé, Université François-Rabelais, Tours, France, 2 Université de Nantes, Faculté d'odontologie, Nantes, France, 3 Odontology Department, Tours University Hospital, Tours, France, 4 Biostatistics and Bioinformatics (DIM), University Hospital, Dijon, France, 5 Bourgogne Franche-Comté University, Dijon, France, 6 Faculté d'Odontologie de Reims, Université Champagne Ardenne, Reims, France, 7 EA 481 Integrative Neurosciences and Clinical, University Hospital of Besançon, Besançon, France, 8 Health Big Data, LTSI—INSERM U 1099, University of Rennes 1, Rennes, France, 9 Rennes University Hospital and Guillaume Regnier Hospital, Rennes, France, 10 INSERM, CIC 1432, Dijon University Hospital, Clinical Investigation Center, clinical epidemiology/ clinical trials unit, Dijon, France, 11 Biostatistics, Biomathematics, Pharmacoepidemiology and Infectious Diseases (B2PHI), INSERM, UVSQ, Institut Pasteur, Université Paris-Saclay, Paris, France

* frederic.denis@chu-tours.fr

**Data Availability Statement:** The database was transmitted by the National Health Insurance Fund for Salaried Workers-CNAMTS (Caisse nationale de l'assurance maladie des travailleurs salaries;

## Abstract

### Objective

To identify the differences between persons with schizophrenia (PWS) and general population in France in terms of oral health treatment (tooth scaling, dental treatment and tooth extraction) and the factors associated with these differences.

### Methods

This retrospective cohort study included PWS identified from a representative sample of 1/97th of the French population (general sample of beneficiaries). PWS were identified from 2014 data by an algorithm that included: F2 diagnostic codes in the register of long-term diseases in 2014 AND {(at least three deliveries of antipsychotics in 2014) OR (F20 diagnostic codes as a main or associated diagnosis in hospital discharge abstracts in 2012 or 2013 (hospital data for medicine, surgery and obstetrics)}. Follow-up dental care was explored for all people over a period of 3 years (2014 to 2017).

### Results

In 2014, 580,219 persons older than 15 years were identified from the 96 metropolitan departments in France; 2,213 were PWS (0.4%). Fewer PWS were found along a diagonal line from north-east to south-west France, and the highest numbers were located in urban departments. PWS were more often male (58.6% vs 48.7%, p<0.001). They were less likely to have had tooth scaling but more likely to have undergone a dental extraction. In one third of departments, more than 50% of PWS had at least one tooth scaling over a three-year period; the rate of dental extraction in these departments ranged from 6 to 23%. Then, a

address: Caisse Nationale de l'Assurance Maladie 50 Avenue du Professeur André Lemierre, 75020 Paris) (responsible for the extraction of EGB data). The use of these data by our department was approved by the National Committee for data protection. Data used in this study are available for researchers who meet the criteria for access to these French data from the National Health Insurance Fund (training that open a personal accreditation, approval of the protocol by required authorities (Expert Committee to researches, studies and evaluations in the health field-CEREES, and National Committee for data protection-CNIL) according to the "Décret n° 2016-1872 du 26 décembre 2016 modifiant le décret n° 2005-1309 du 20 octobre 2005 pris pour l'application de la loi n° 78-17 du 6 janvier 1978 relative à l'informatique, aux fichiers et aux libertés", https://www.legifrance.gouv.fr/eli/decret/2016/12/26/2016-1872/jo/texte. A permanent access is granted to some public services within the limits established by laws (Décret n° 2016-1871 du 26 décembre 2016 relatif au traitement de données à caractère personnel dénommé « système national des données de santé », https://www.legifrance.gouv.fr/eli/decret/2016/12/26/2016-1871/jo/texte. Contact: Caisse Nationale de l'Assurance Maladie, 50 Avenue du Professeur André Lemierre, 75020 Paris

**Funding:** The authors received no specific funding for this work

**Competing interests:** The authors have declared that no competing interests exist.

quarter of the departments in which 40 to 100% of PWS had had at least one dental extraction (2/8) presented a rate of tooth scaling ranging from 0 to 28% over the study period.

## Conclusions

Compared with the general population, PWS were less likely to have had tooth scaling and dental treatment but more likely to have undergone dental extraction.

## Introduction

Schizophrenia affects between 0.7 and 1% of the worldwide population and 400 000 to 600 000 individuals in France [1]. This mental illness is a severe psychiatric disorder characterized by significant cognitive and emotional disruptions. Schizophrenia requires long-term medical treatment, which can result in physical, psychological, and social problems related to both the disease and the potential side effects of treatment [2]. The negative symptoms of schizophrenia, such as lack of initiation, a lack of concern for personal health, social withdrawal and a lack of motivation, may lead persons with schizophrenia (PWS) to neglect their self-care resulting in higher rates of physical ill-health [3,4]. Studies show that 19 to 57% of PWS have at least one associated somatic condition, including cardiovascular, gastrointestinal, respiratory, neoplastic, infectious, endocrine, and oral disorders. Furthermore, these persons suffer from stigmatization and inequities in terms of access to treatment, and thus about half of these comorbidities remain undiagnosed [5–8]. Excess mortality rates due to the complications of a chronic physical illness are two or three times higher in such patients than in the general population, resulting in a 10 to 25-year reduction in life expectancy in comparison with the rest of the population [9–10].

The trend towards poor physical health in people with mental illness has been the subject of growing attention [11,12], but there has been less interest in the issue of oral health [13–15]. Nevertheless, oral health is an important part of overall physical condition. In their original review on this topic, Kisely et al. reported the reasons for poor oral health in persons with severe mental illness, including PWS, and its impact on general health [16]. Specifically, dental caries and periodontal measurement indexes often reach twice the level found in the general population [17–19]. A number of combined factors contribute to the poor oral health of these individuals. These factors can include orales infectious diseases interacting with metabolic disturbances induced by antipsychotic treatments (diabetes, obesity), and also poor diet and lifestyle behaviors (high-sugar diet, use of psychoactive substances such as tobacco, and inadequate oral hygiene) [20–22]. Finally, poor oral health impacts social functioning and affects quality of life, self-esteem and self-confidence, which are already dramatically lower in this population [16,23].

In the French department of Côte d'Or (530,000 inhabitants), a previous study was conducted with PWS, using a random stratified method. This study showed that, compared with data available for the general population, there were more extractions and missing teeth and fewer dental fillings [24]. But this study was limited to one area of France and was based on data from volunteers, who may differ from the target population. Furthermore, the most recent available data on oral health epidemiology and for the general population are more than 20 years old [25].

Our aim was to identify the differences between PWS and general population in France in terms of oral health treatment (tooth scaling, dental treatment and tooth extraction) and the factors associated with these differences.

## Materials and methods

### Design of the study and sample of the population

This retrospective cohort study focused on PWS identified from a representative sample of 1/97[th] of the French population that is known as the EGB (*echantillon généraliste de bénéficiaires*, general sample of beneficiaries). Follow-up dental care was explored for all people in the EGB over a period of 3 years (2014 to 2017).

The EGB was constructed on a national level by the French health insurance agency, which manages the representativeness of the data. It was drawn randomly from a check digit of the beneficiary's identification number. From a stable, representative sample of the French population, these data allowed us to estimate care pathways and follow-up while excluding the effects of the geographical area, health facility or practices. The EGB was based on the "National System of Health Data" (SNDS) database which collects individual hospital and non-hospital healthcare data. These data include automatically recorded healthcare acts such as biological testing, treatment, medical transport and a register for long-term diseases. Health insurance is compulsory for everyone living in France, and each health act is reimbursed by the French health insurance agency and therefore recorded in SNDS. These data include all types of mandatory health insurance systems (the main national health insurance, health insurance for the agricultural sector, health insurance for the self-employed and 12 other specific health insurance schemes) covering more than 90% of the French population. In 2016, the EGB sample was made up of almost 600,000 health insurance beneficiaries. The reliability of the SNDS has been established in recent studies [26,27].

### Groups constitution

**Persons with schizophrenia identification.**   PWS were identified for the year 2014 by an algorithm that included:

- Diagnostic Codes F2 in the register of long-term diseases [28] in 2014 AND {(at least three deliveries of antipsychotics in 2014) OR (Diagnostic codes F20 as a main or associated diagnosis in hospital discharge abstracts in 2012 or 2013 (hospital data for medicine, surgery and obstetrics)}

The purpose of this algorithm was to identify adult schizophrenia in the SNDS database. It was built using the information obtained from interviews with experts in schizophrenia and based on their procedures to identify in- or out-patients [29].

**Control identification.**   The control group was composed of all the people who are included in the EGB and not included in the cohort of patients with schizophrenia.

Because the diagnosis of schizophrenia is rare and difficult before age 15 years [30], we only recruited individuals over 15 years old.

### Outcomes

**Variables.**   The main outcomes of interest (in the French Common Classification of Procedures) were; 1) tooth scaling (code HBJD001); 2) dental treatments (codes HBMD0-, HBFD0-); and 3) tooth extraction (code HBGD0-) [31]. These 3 outcomes are covered by French National Insurance.

1. Tooth scaling is a common dental cleaning procedure to remove plaque buildup. Tooth scaling prevents periodontal disease. It is the most common procedure in general dentistry.

2. Dental treatments were all procedures for filling cavities, root canal treatments, such as exeresis of canal content, or exeresis of pulp, tooth restoration, and denture-repair.

3. Tooth extractions were identified by their location in the mouth, i.e. avulsion of canine tooth, ectopic tooth, or molar tooth.

The outcomes of interest were explored as qualitative variables: binary (at least one dental treatment) or nominal variables with more than 2 categories (1, 2–3 or >3).

The explanatory variables were age and gender; these variables were assessed in the group of patients with schizophrenia and the group without schizophrenia.

## Statistical analysis

Qualitative variables were expressed as percentages and were first compared between the two groups with and without schizophrenia using the Fisher exact test, under the conditions of application. The number and percentage of dental acts weres presented by class (1, 2–3, or >3) for each type of care.

**Geographic analysis.** The geographic scale used for this analysis was the geographic department code recorded in the EGB. France is divided into 96 metropolitan departments, with populations ranging from 77,000 to 2,577,000 inhabitants.

**Mapping of population and distribution of dental care.** For each department of residence, the size of the symbol varied proportionally with the number of people with schizophrenia who resided there. Access to each type of dental care was calculated by plotting the number of overall PWS against the number of patients receiving dental care. The rate of dental care was expressed as a percentage for each category.

**Multivariate logistic regressions analyses.** To estimate the association between schizophrenia and dental outcomes, multivariate logistic regressions and multivariate logistic regressions adjusted for sex were performed by age category (15–24, 25–34, 35–44, 45–54, 55–64, ≥65). The results are reported as adjusted odds ratios (aOR) with 95% confidence intervals (CI).

Two sensitivity analyses were performed for other algorithms.

First, we identified PWS from an algorithm used by the National Institute for Health Surveillance (InVS) [32] in order to estimate the prevalence of schizophrenia in France:

- F20 diagnostic codes in the register of long-term diseases associated with the F20 codes as the main or associated diagnoses in discharge abstracts (hospital data for medicine, surgery and obstetrics and/or psychiatry)

- And/or at least three annual deliveries of antipsychotics in 2014 associated with the F20 ICD-10 codes as the main or associated diagnoses in discharge abstracts (hospital data for medicine, surgery and obstetrics and/or psychiatry) over the past four years.

Our algorithm did not include data from psychiatric hospitals.
Second, we identified PWS with a less restrictive algorithm, including the presence of

- one of the F20 diagnostic codes in the register for long-term diseases,

- and/or one of the F20 diagnostic codes as the main or associated diagnoses in discharge abstracts (hospital data for medicine, surgery and obstetrics),

- and/or at least three annual deliveries of antipsychotics in 2014.

A p-value of 0.05 was set to define statistical significance for all analyses. SAS 9.3 software was used for data analyses. The geographic information system MapInfo 11.0 was used for mapping.

### Ethical approval

French university hospital researchers have a permanent authorization to manage data from the EGB as indicated in the Decree n˚ 2016–1871 of December 26, 2016 on processing of personal data from SNDS. Data was treated by individuals who were authorized by the State. This study was conducted in accordance with the Declaration of Helsinki. Individual written consent was not needed for this study.

## Results

### Population and distribution of dental care

In 2014, 580,219 persons over 15 years old were identified in EGB. Among them, 2,213 PWS (0.4%) were identified by our algorithm.

Every department in metropolitan France presented at least one case of schizophrenia in 2014. The departments located along a diagonal line from the north-east to the south-west had fewer cases of schizophrenia, while the highest numbers were in urban departments, those bordering the Mediterranean, and along the external borders (Fig 1).

Our comparison found a significant difference between PWS and persons without schizophrenia in terms of age and gender (Table 1).

PWS were more often male (58.6% vs 48.7%, p<0.001) and more likely to be aged 35 to 64 years.

PWS were less likely to have tooth scaling and more to have dental extraction during the 3-years follow-up period. The overall rate of tooth scaling was 41.5% for PWS and 48.0% for people without schizophrenia (p<0.0001). In PWS, the rate of tooth scaling decreased with age. There was a significant difference in the frequency of tooth extraction between the two groups: 22.7% of PWS had had at least one dental extraction versus 18.4% of people without schizophrenia (p<0.0001). The spatial distribution of tooth scaling (Fig 2), dental treatment (Fig 3) and tooth extraction (Fig 4) appeared to be random. In one third of departments, more than 50% of PWS had at least one recorded tooth scaling over the three-year period.

In these same departments, the rate of dental extractions ranged from 6 to 23%. In eight departments, at least one dental extraction was recorded for 40 to 100% of PWS. A quarter of these departments (2/8) reported a rate of tooth scaling ranging from 0 to 28% over the study period.

### Multivariate logistic regressions analysis

The results of adjusted logistic regression analyses within 3 years are presented in Table 2.

Between 15 and 44 years, schizophrenia was not significantly associated with tooth scaling after adjustment for sex. Regression logistic analysis showed that schizophrenia was associated with decreased frequency of tooth scaling from the age of 45 years (aOR = 0.45 95% CI [0.37–0.55] for age 55–64 years). Schizophrenia was significantly associated with a higher frequency of dental treatment for 25–34 year olds (aOR = 1.42[1.15–1.76]) and a decreased frequency of dental treatment after 55 years (aOR = 0.63 95% CI [0.52–0.77] for age 55–64 years). Then, schizophrenia was associated with an increased risk of dental extraction in patients 15 to 54 years old. The presence of schizophrenia yielded an aOR of 2.05 (95% CI 1.18–3.57) for youths aged 15 to 24. After 55 years, the risk of dental extraction was not significant. Female gender was associated with more frequent tooth scaling and dental treatment, regardless of age.

## Discussion

To our knowledge, this study is the first to use national data to analyze the oral health treatment habits of PWS based on a large representative sample (2,213).

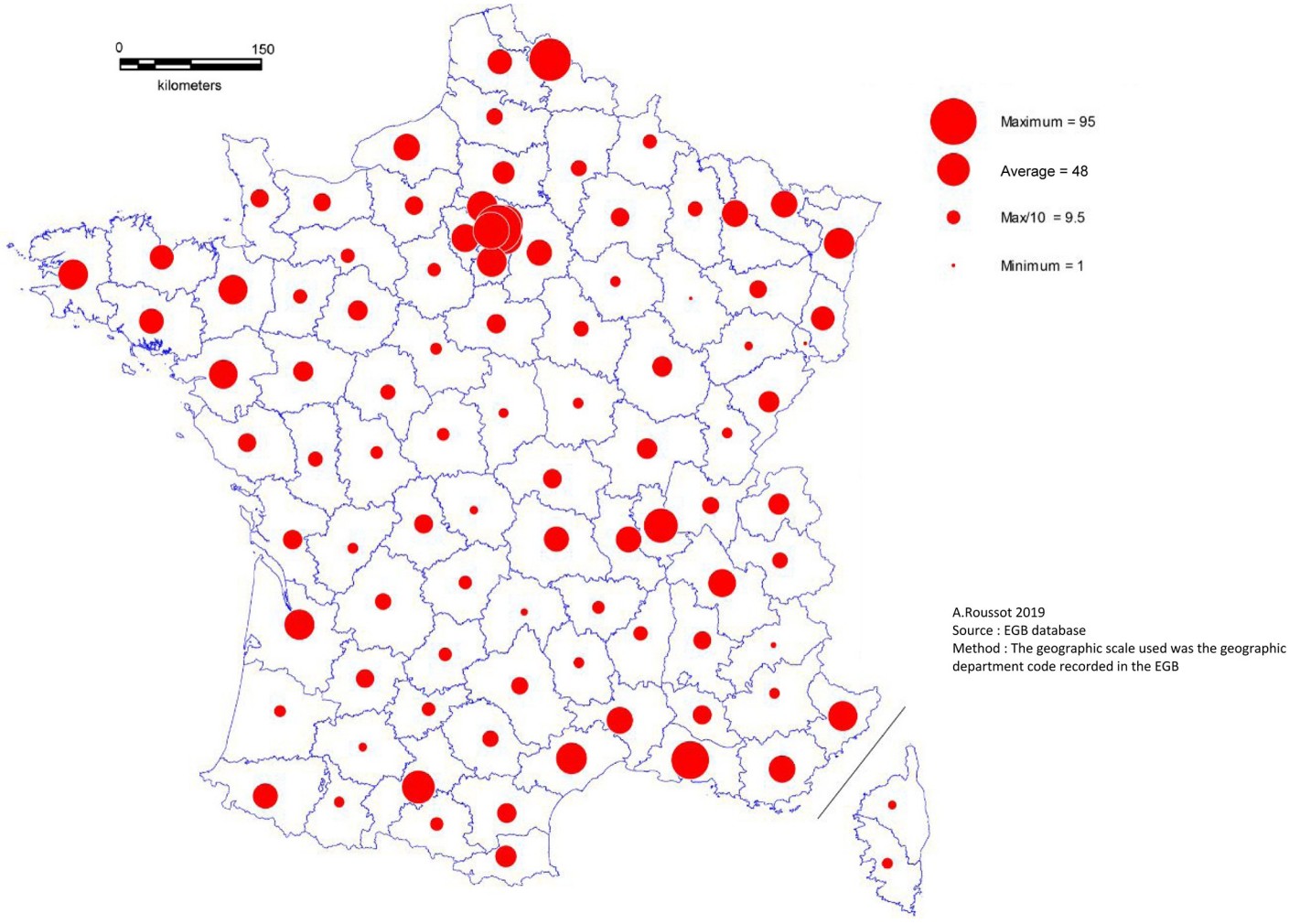

**Fig 1. Geographical distribution of persons with schizophrenia.**

We highlighted certain disparities between the oral health treatment trends for PWS and for the general population. These disparities include both the type and frequency of care on a national scale.

It is well known that regular professional tooth cleaning is important for the prevention of dental caries and periodontal disease [33]., In addition, though the "Haute Autorité de Santé" suggested regular monitoring of oral health in 2010 [34], annual tooth scaling has only been recommended in France since 2013–2014 [35]. PWS seem to have more difficulty accessing the benefits provided bythe National Health Insurance program, leading to inadequate dental care (less tooth scaling and more dental extraction) [13,24,36]. This could explain why PWS access oral health care less frequently and have worse dental health than the general population.

The social and psychological impact of dental extractions can have significant consequences on quality of life [13,37]. The basic ability to chew food can also be affected. Tooth Extraction may be a quick-fix solution to pain and discomfort but there are benefits to saving and restoring dentition [37–39].

Table 1. Population and distribution of dental care.

| | With schizophrenia (n = 2.213) % | | Without schizophrenia (n = 578,006) % | | p-value |
|---|---|---|---|---|---|
| Age | | | | | |
| 15–24 | 74 | 3.3 | 87,994 | 15.2 | |
| 25–34 | 339 | 15.3 | 96,254 | 16.7 | |
| 35–44 | 572 | 25.9 | 97,262 | 16.8 | <0.001 |
| 45–54 | 525 | 23.7 | 97,291 | 16.8 | |
| 55–64 | 428 | 19.3 | 85,459 | 14.8 | |
| ≥65 | 275 | 12.4 | 113,746 | 19.7 | |
| Gender | | | | | |
| Male | 1,297 | 58.6 | 281,434 | 48.7 | <0.001 |
| Female | 916 | 41.4 | 296,572 | 51.3 | |
| Tooth scaling | 919 | 41.5 | 277,206 | 48.0 | <0.001 |
| 1 | 280 | 12.7 | 90,152 | 15.6 | |
| 2–3 | 415 | 18.8 | 126,494 | 21.9 | <0.001 |
| >3 | 223 | 10.1 | 60,560 | 10.5 | |
| Dental treatments | 927 | 41.9 | 242,809 | 42.0 | 0.91 |
| 1 | 194 | 8.8 | 67,973 | 11.8 | |
| 2–3 | 323 | 14.6 | 84,637 | 14.6 | <0.001 |
| >3 | 410 | 18.5 | 90,199 | 15.6 | |
| Tooth extraction | 503 | 22.7 | 106,108 | 18.4 | <0.001 |
| 1 | 307 | 13.9 | 71,359 | 12.4 | |
| 2–3 | 154 | 7.0 | 30,330 | 5.2 | <0.001 |
| >3 | 42 | 1.9 | 4.419 | 0.8 | |

*Fisher exact test

Compared with other types of medical care, dental visits are widely underutilized by PWS, which means that dental treatment should be delivered in a more preventive manner [40]. The disparities in preventive care between PWS and the general population may also be due to the difficulties faced by PWS when accessing and using health services [39,40]. For instance, dentists are more likely consider these patients as "difficult to manage", which may lead to them to opt for simple tooth extractions, foregoing the more complicated methods. Such practices or behavior may result in cases of "early missing teeth" [41]. Schizophrenia may severely interfere with patients seeking dental treatment, delaying restorative treatment until tooth loss is inevitable. Moreover, studies have shown that dental health is often seen as lower priority in the global health approach of PWS [42,43].

The results of the multivariate logistic regressions analysis shows that PWS who are male or older were especially likely to have less tooth scaling and more dental extraction. On the contrary, women generally had more positive oral health attitudes and practices (using extra cleaning devices for example) [44,45]. In a population of institutionalized psychiatric patients, Ngo DY et al. [46] also found that males had more tooth decay than females. Compared with male patients, female patients may perceive a decayed tooth to be more problematic and report symptoms, meaning that they are more likely obtain dental care [46]. In an American study, dental health was linked to sex and socioeconomic status, and to a lesser extent to education and income, whereas age, race and dental insurance coverage were not significant [45]. Put simply, women are more likely than men to take care of themselves in daily life. Men therefore require additional encouragement to attend dental consultations.

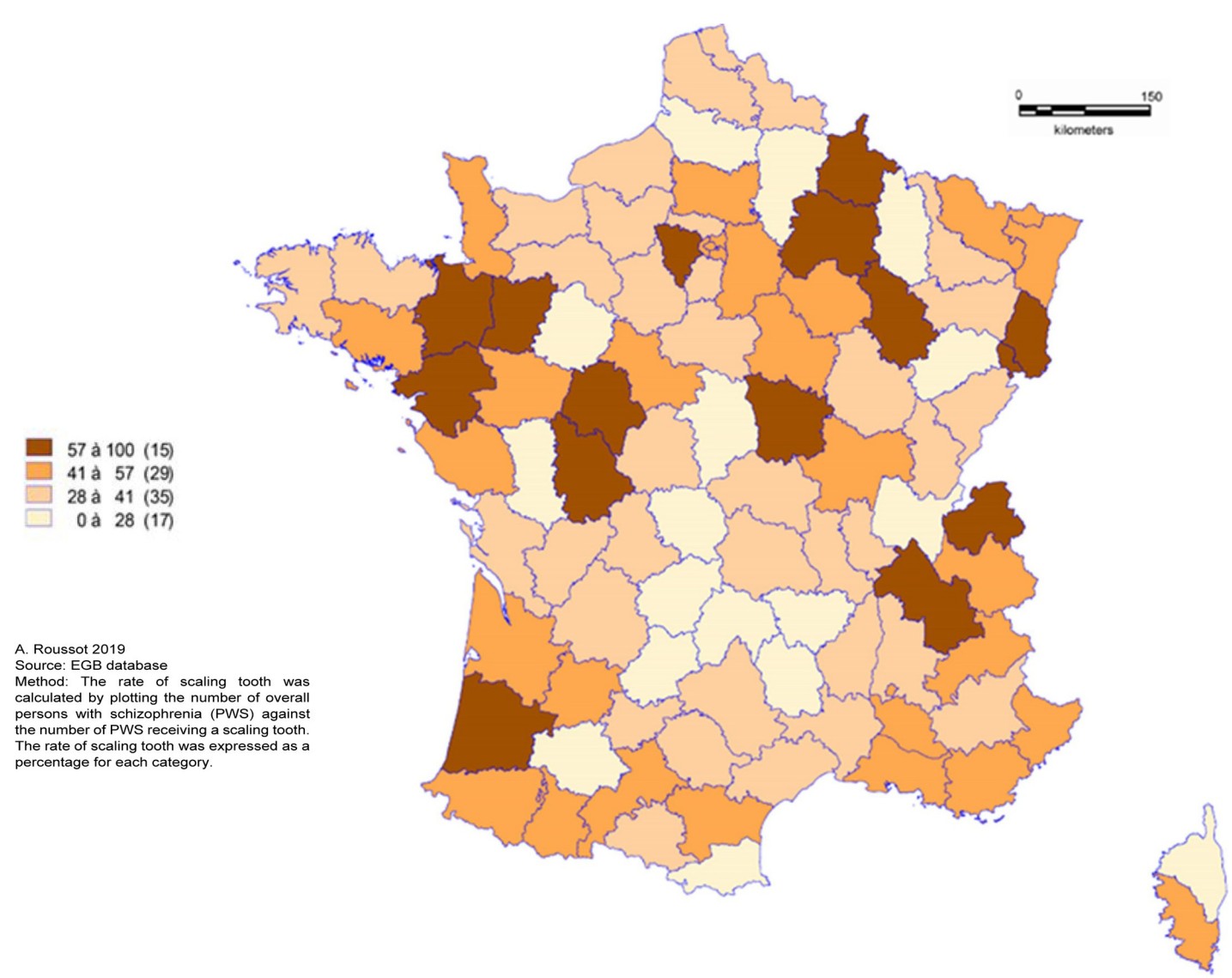

A. Roussot 2019
Source: EGB database
Method: The rate of scaling tooth was calculated by plotting the number of overall persons with schizophrenia (PWS) against the number of PWS receiving a scaling tooth. The rate of scaling tooth was expressed as a percentage for each category.

**Fig 2. Geographical distribution of scaling tooth rate for persons with schizophrenia.**

Studies that have investigated age as contributing factor in tooth scaling or dental treatment are contradictory. Advanced age contributed to less frequent tooth brushing for Harada et al. [44], but other authors found no significant association [43]. Though elderly people often lose teeth as a result of periodontal disease and caries and there is an increase in tooth extractions with age, other factors such as education level and economic status are worth taking into account [47].

Recent international studies have showed that psychiatric inpatients lack sufficient preventive oral health care [3,13–20], but this is the first study to highlight a gap in oral health treatment in PWS in France, and we describe the obvious disparity in consumption that underlies poor oral health. These results also appear to be generally compatible with reports from the literature related to the oral health of PWS [15,19,34,43,46,47]. Dental ailments are largely preventable, and preventive measures such as daily tooth brushing and regular dental checkups are simple and easy to implement [48]. In contrast, the combination of poor diet and lifestyle

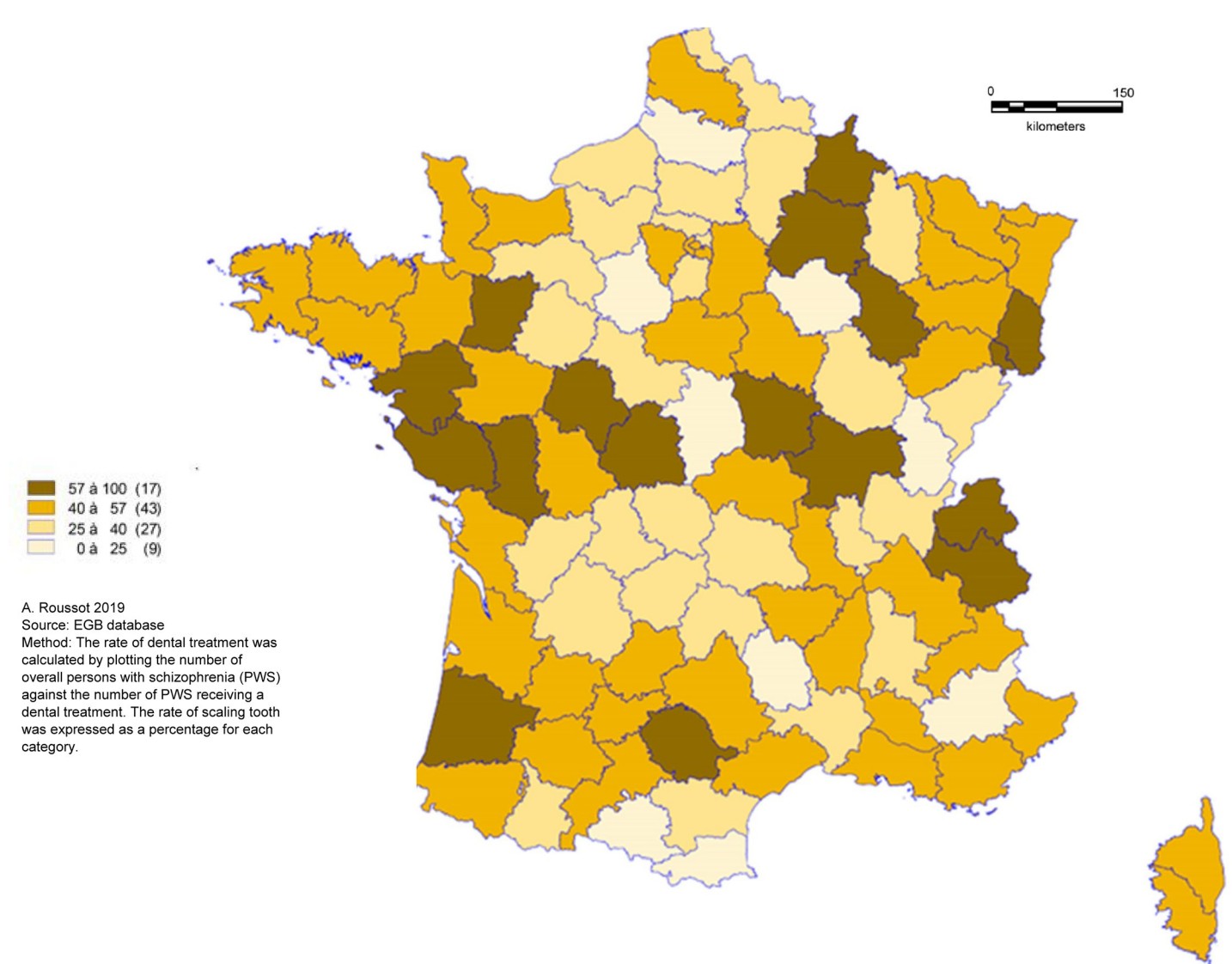

57 à 100  (17)
40 à  57  (43)
25 à  40  (27)
 0 à  25   (9)

A. Roussot 2019
Source: EGB database
Method: The rate of dental treatment was
calculated by plotting the number of
overall persons with schizophrenia (PWS)
against the number of PWS receiving a
dental treatment. The rate of scaling tooth
was expressed as a percentage for each
category.

**Fig 3. Geographical distribution of dental treatment rate for persons with schizophrenia.**

behaviours (high-sugar diet, use of psychoactive substances such as tobacco, and inadequate oral hygiene lead to poor health [3,16,17]. The future development, testing and implementation of interventions to improve oral health in PWS can be guided by the present findings, which provide important data regarding shortcomings in treatment and the need for more regular dental visits in this population.

## Limitation and strengths

The algorithm used to identify PWS did not include the database of psychiatric hospitalizations even though it was included in the reference algorithm. This is because the EGB does not include the French psychiatric database. However, the persons who were covered as part of long-term disease with diagnostic codes F20- represent approximately 70% of people identified by the SNDS as suffering from psychotic disorders.

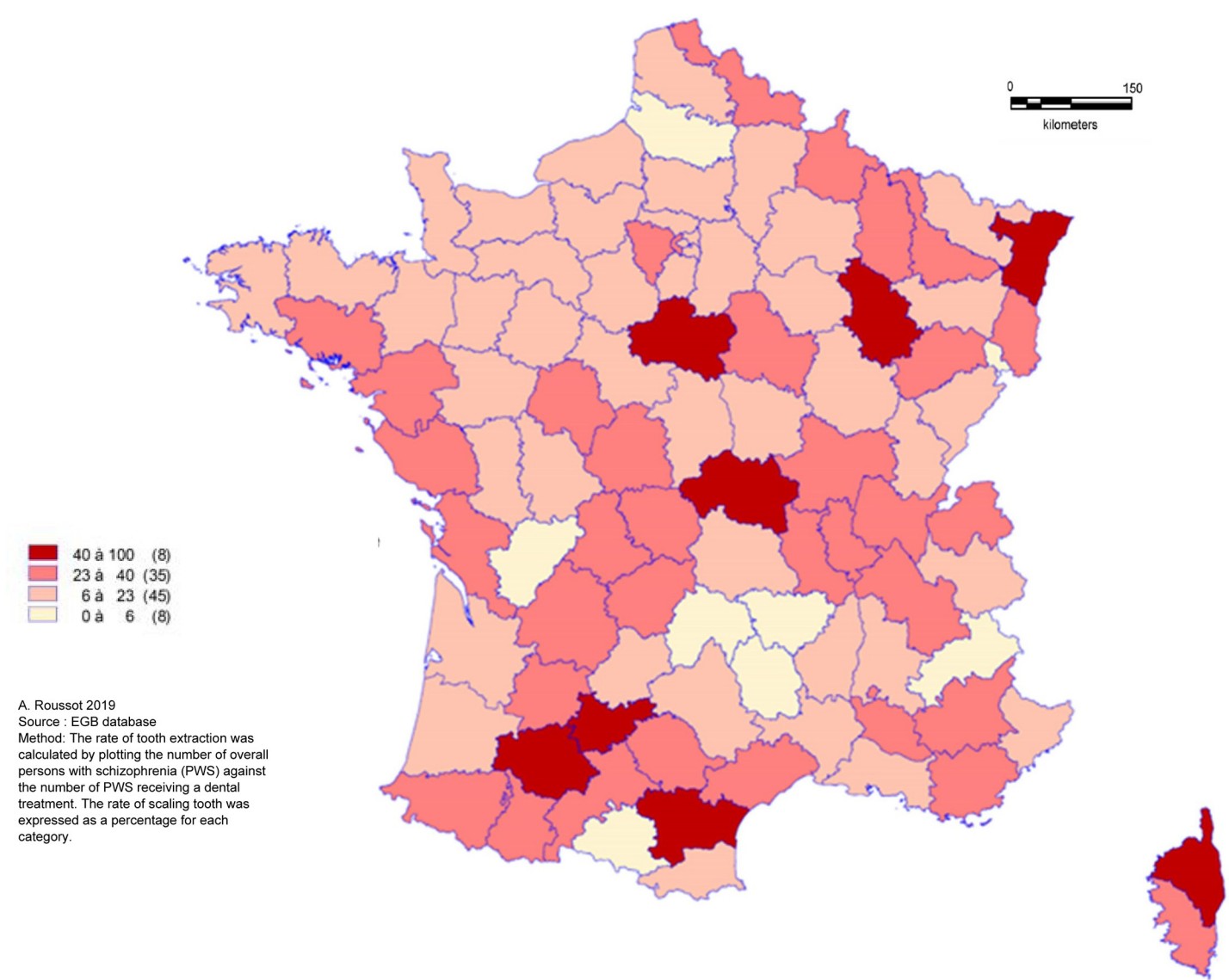

Legend:
40 à 100 (8)
23 à 40 (35)
6 à 23 (45)
0 à 6 (8)

A. Roussot 2019
Source : EGB database
Method: The rate of tooth extraction was calculated by plotting the number of overall persons with schizophrenia (PWS) against the number of PWS receiving a dental treatment. The rate of scaling tooth was expressed as a percentage for each category.

**Fig 4. Geographical distribution of tooth extraction rate for persons with schizophrenia.**

Furthermore, this bias is probably of limited effect. The potential consequence would be an inclusion of unidentified cases of schizophrenia in the control group, therefore reducing the amplitude of the reported associations.

One of the strengths of our study is the representativeness of the EGB of the French general population and its high stability over time. An additional strength is the fact that health insurance is compulsory for all French residents. This means that all health care acts are reimbursed, and all reimbursements are automatically recorded and transferred to the SNDS database. The quantification of dental care can therefore be considered exhaustive.

Moreover, our results suggest a spatial prevalence of schizophrenia that is tied to urbanicity. Urbanicity is a well-established environmental risk factor for developing schizophrenia [49], and the prevalence of schizophrenia is known to be greater in cities than in rural areas [49,50].

In accordance with a number of published studies, our PWS sample was predominantly male [51]. However, patients were more likely to be aged 35 to 64 even though the initial

**Table 2. Multivariate logistic regressions analysis.**

| Age category | | 15–24 | | | 25–34 | | | 35–44 | |
|---|---|---|---|---|---|---|---|---|---|
| | OR | IC 95% | p | OR | IC 95% | p | OR | IC 95% | p |
| Tooth scaling | | | | | | | | | |
| CG | 1.05 | 0.65–1.67 | 0.85 | 1.24 | 0.99–1.54 | 0.05 | 0.95 | 0.80–1.11 | 0.5 |
| Gender | 1.45 | 1.41–1.49 | <0.0001 | 1.66 | 1.62–1.70 | <0.0001 | 1.53 | 1.49–1.57 | <0.0001 |
| Dental extraction | | | | | | | | | |
| CG | 2.05 | 1.18–3.57 | 0.01 | 1.82 | 1.40–2.37 | <0.0001 | 1.53 | 1.26–1.87 | <0.0001 |
| Gender | 1.31 | 1.26–1.37 | <0.0001 | 1.04 | 101–1.08 | 0.04 | 1.01 | 0.97–1.04 | 0.75 |
| Dental treatment | | | | | | | | | |
| CG | 1.34 | 0.83–2.17 | 0.22 | 1.42 | 1.15–1.76 | 0.001 | 1.11 | 0.94–1.31 | 0.21 |
| Gender | 1.24 | 1.20–1.27 | <0.0001 | 1.30 | 1.27–1.33 | <0.0001 | 1.30 | 1.27–1.34 | <0.0001 |
| Age category | | 45–54 | | | 55–64 | | | ≥65 | |
| | OR | IC 95% | p | OR | IC 95% | p | OR | IC 95% | p |
| Tooth scaling | | | | | | | | | |
| CG | 0.63 | 0.53–0.75 | <0.0001 | 0.45 | 0.37–0.55 | <0.0001 | 0.42 | 0.32–0.57 | <0.0001 |
| Gender | 1.52 | 1.48–1.56 | <0.0001 | 1.48 | 1.44–1.52 | <0.0001 | 1.17 | 1.14–1.19 | <0.0001 |
| Dental extraction | | | | | | | | | |
| CG | 1.39 | 1.15–1.69 | 0.001 | 0.87 | 0.70–1.10 | 0.25 | 0.82 | 0.61–1.11 | 0.19 |
| Gender | 0.97 | 0.94–1.00 | 0.05 | 0.98 | 0.95–1.01 | 0.11 | 0.93 | 0.90–0.95 | <0.0001 |
| Dental treatment | | | | | | | | | |
| CG | 0.84 | 0.71–1.00 | 0.05 | 0.63 | 0.52–0.77 | <0.0001 | 0.7 | 0.54–0.89 | 0.005 |
| Gender | 1.26 | 1.23–1.30 | <0.0001 | 1.23 | 1.20–1.23 | <0.0001 | 1.06 | 1.03–1.08 | <0.0001 |

OR: odds radio; CI: Confidence Interval; CG: Control group; Gender (ref = male)

symptoms of schizophrenia often begin before 20 years of age [52]. It is possible that mental illness was often undiagnosed in the early stages or that the algorithm used in our study was not accurate enough. It is known that untreated psychosis in first-episode cases means that patients are often actively psychotic for a very long time before they get help [53]. For this reason, we cannot rule out a number of unidentified PWS in our study.

## Conclusion

This is the first study to provide geographic mapping of the oral health treatment of PWS in France. We found that the departments located along a north-east/south-west diagonal line had fewer PWS while the highest number were in urban departments, those bordering the Mediterranean, and along the external borders. We highlighted that PWS were less likely to have had tooth scaling and dental treatments but more likely to have undergone a dental extraction than general population. We noticed a clear inequity in oral health treatment, and national health policies are needed to address this issue. Further studies are needed in order to improve the management of oral health in PWS, potentially with the use of specific prevention and education programs. But first of all, the most pressing challenge is to expand the understanding of the factors that limit or facilitate the healthcare pathway for PWS in order to optimize their oral health.

## Acknowledgments

The authors are grateful to Suzanne Rankin for revising the manuscript

## Author Contributions

**Conceptualization:** Frédéric Denis, Karine Goueslard, Catherine Quantin.

**Data curation:** Frédéric Denis, Karine Goueslard.

**Formal analysis:** Frédéric Denis, Karine Goueslard, Catherine Quantin.

**Investigation:** Frédéric Denis.

**Methodology:** Frédéric Denis, Valérie Bertaud, Catherine Quantin.

**Project administration:** Frédéric Denis, Catherine Quantin.

**Resources:** Karine Goueslard.

**Software:** Karine Goueslard.

**Supervision:** Frédéric Denis, Francesca Siu-Paredes, Gilles Amador, Emmanuel Rusch, Valérie Bertaud, Catherine Quantin.

**Validation:** Frédéric Denis.

**Visualization:** Frédéric Denis, Francesca Siu-Paredes, Gilles Amador, Emmanuel Rusch, Valérie Bertaud.

**Writing – original draft:** Frédéric Denis, Karine Goueslard, Catherine Quantin.

**Writing – review & editing:** Frédéric Denis, Karine Goueslard, Catherine Quantin.

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
