## [Decision Letter · Decision Letter 0]

3 Jan 2020

PONE-D-19-20482

Oral health consumption habits of people with schizophrenia in France: a retrospective cohort study

PLOS ONE

Dear Dr. Denis,

Thank you for submitting your manuscript to PLOS ONE. After careful consideration, we feel that it has merit but does not fully meet PLOS ONE’s publication criteria as it currently stands. Therefore, we invite you to submit a revised version of the manuscript that addresses the points raised during the review process.

We would appreciate receiving your revised manuscript by Jan 23 2020 11:59PM. To enhance the reproducibility of your results, we recommend that if applicable you deposit your laboratory protocols in protocols.io, where a protocol can be assigned its own identifier (DOI) such that it can be cited independently in the future. For instructions see: http://journals.plos.org/plosone/s/submission-guidelines#loc-laboratory-protocols

We look forward to receiving your revised manuscript.

Kind regards,

Denis Bourgeois

Academic Editor

PLOS ONE

Journal Requirements:

1. Please include your tables as part of your main manuscript and remove the individual files. Please note that supplementary tables (should remain/ be uploaded) as separate "supporting information" files

Reviewers' comments:

Reviewer's Responses to Questions

**Comments to the Author**

1. Is the manuscript technically sound, and do the data support the conclusions?

Reviewer #1: Partly

Reviewer #2: Yes

2. Has the statistical analysis been performed appropriately and rigorously? 

Reviewer #1: I Don't Know

Reviewer #2: Yes

3. Have the authors made all data underlying the findings in their manuscript fully available?

Reviewer #1: Yes

Reviewer #2: Yes

4. Is the manuscript presented in an intelligible fashion and written in standard English?

Reviewer #1: Yes

Reviewer #2: Yes

5. Review Comments to the Author

Reviewer #1: This submission is about "Oral health habits of people with schizophrenia in Franc"e. This is an analysis of a database dating back to 2014 and which may be considered as irrelevant.

The interest in an international journal is limited as shown in Figures 1-4.

The methodology is not very detailed. Under the term "oral health consumption", the authors select care variables that are not sufficiently descriptive of the accessible dental care panel. Example: "dental treatments"

The discussion is vague and does not concretely answer the objective

The references are not updated and very targeted on France.

Reviewer #2: * This article is well-written and interesting, bringing new knowledge in the epidemiological field of oral health among persons with schizophrenia.

* Title and objective. The term "consumption" might not be relevant for the present study, given that what is measured is more than just "the using up of goods and services". As explained in the introduction, PWS can have difficulties to access to treatment (which is not directly linked to the consumption). Maybe the term "treatment" would be more relevant.

* Introduction section.

- P3, l22 "These factors can include dental caries, periodontal diseases" is unnecessary, since explained factors are related to "dental caries and periodontal measurement indexes (l20).

- P3, l28-31. Please indicate in the text that ref 23 is about PWS.

- P4, l32-33: instead of "on this subject", indicate "on oral health epidemiology".

* Material and methods

- P4, l57: The paragraph should be entitled "Groups constitution", with two clear sub-paragraphs: PWS identification and control identification.

- P5, l70-1: The sentence "Follow-up dental care was explored for all people in the EGB over a period of 3 years (2014 to 2017)" should be placed in the "design of the study and sample of the population" section, because it allows to understand why the study is a retrospective cohort.

-P5, l76: please indicate that these 3 outcomes are covered by French National Insurance

-P5, l81: "The outcomes of interest were explored as qualitative variable" should rather be ""The outcomes of interest were explored as binary (at least one dental treatment) and ordinal qualitative (1,2-3 or >3) variables." Please make the changes in all the manuscript (e.g. p5, l86, etc)

- P5, l87: the Fisher exact test is OK. Pearson χ2 test is not necessary.

- P5, l88: The number and percentage of dental acts were presented ...

- P5-6: "First, PWS were localized according to their place of residence. However, mapping by residence may have compromised patient anonymity in a few cases, so we chose to map according to the department of residence." is OK, but should be removed in the text because it doesn't add meaningful information.

- P6, l96: Paragraph should be entitled "Mapping of population and distribution of dental care" , thus "We mapped the distribution of PWS and the distribution of dental care" l97 should be removed.

- P6, l102: As it is an observational study, authors should be cautious with causation. Thus, it should be written something like " To estimate the association between schizophrenia and dental outcomes, etc..."

*Results

Results are well reported

p7, l137: "confirmed" is not necessary

* Discussion

Discussion is well conducted

-p8, l180, typo "carries"

- Maybe the paragraph p8, l171-177 should appear in the limitation section of the discussion.

Figures

in the legends, please change "à" to "to"

Table 1: Presentation of p-values should be clearer. Readers should easily understand that there are two types of tests in the table (for binary outcomes and qualitative ordinal outcomes)

Table 2: Authors should keep the same definitions in the text and in the table. The labeling of "Persons without schizophrenia" is unclear.

6. PLOS authors have the option to publish the peer review history of their article (what does this mean?). If published, this will include your full peer review and any attached files.

Reviewer #1: No

Reviewer #2: No

---

## [Author Response · Author response to Decision Letter 0]

17 Feb 2020

Response to reviewers

Reviewer #1: 

We thank the Reviewer for giving us an opportunity to substantially improve the content and the presentation of our manuscript. We have modified the article in accordance with your requests. You will find every modification in the text using track changes, and the pages are noted in the answer for every point below. We hope we have met your requirements to improve this paper.

This submission is about "Oral health habits of people with schizophrenia in France". This is an analysis of a database dating back to 2014 and which may be considered as irrelevant. 

Response: Please consider that this retrospective study explored a follow-up dental care for a period of 3 years (2014 to 2017) for persons with schizophrenia.

The interest in an international journal is limited as shown in Figures 1-4.

The methodology is not very detailed. Under the term "oral health consumption", the authors select care variables that are not sufficiently descriptive of the accessible dental care panel. Example: "dental treatments"

Response: 

We thank the Reviewer for this comment. We completed the definition of the procedures and added some examples. 

In Methods section, page 5, lines 84-87:

2) Dental treatments were all procedures for filling cavities, root canal treatments, such as exeresis of canal content or exeresis of pulp, tooth restoration, and denture repair. 

3) Tooth extractions were identified by their location in the mouth, i.e. avulsion of canine tooth, ectopic tooth, or molar tooth.

The discussion is vague and does not concretely answer the objective

Response: We agree with the Reviewer that the discussion needed to be revised. In particular, the first paragraph was rather vague and did not provide the main results of our paper regarding our objective. This paragraph was rewritten accordingly. 

The references are not updated and very targeted on France.

Response: We thank the reviewer for this relevant comment. 

We agree that our references were very targeted on France. In fact, the most recent available data on this subject on oral health epidemiology in France for the general population are more than 20 years old, except for the following reference that we included in our paper.

[15]-Bertaud-Gounot V, Kovess-Masfety V, Perrus C, Trohel G, Richard F. Oral health status and treatment needs among psychiatric inpatients in Rennes, France: a cross-sectional study. BMC Psychiatry 2013, 13:227

We also included references regarding other countries:

[46]-Di Ying Joanna Ngo, W. Murray Thomson, Mythily Subramaniam, Edimansyah Abdin,Kok-Yang Ang. The oral health of long-term psychiatric inpatients in Singapore. Psychiatry Research. 2018; 266:206-11.

[43]-Velasco‑Ortega E, Monsalve‑Guil L, Ortiz‑Garcia I, Jimenez‑Guerra A, Lopez‑Lopez J, Segura‑Egea J.J. Dental caries status of patients with schizophrenia in Seville, Spain: a case–control study. BMC Res Notes (2017) 10:50

[34]-Chu, K.-Y., Yang, N.-P., Chou, P., Chiu, H.-J., Chi, L.-Y., 2012. Comparison of oral health between

inpatients with schizophrenia and disabled people or the general population. J. Formos. Med. Assoc. 111 (4), 214–219.

[19]-Ramon, T., Grinshpoon, A., Zusman, S., Weizman, A., 2003. Oral health and treatment

needs of institutionalized chronic psychiatric patients in Israel. Eur. Psychiatry 18 (3), 101–105.

Reviewer #2: 

1-This article is well-written and interesting, bringing new knowledge in the epidemiological field of oral health among persons with schizophrenia.

Response: We thank the Reviewer for giving us an opportunity to substantially improve the content and the presentation of our manuscript. We have modified the article in accordance with your requests. You will find every modification in the text using track changes, and the pages are noted in the answer for every point below. We hope we have met your requirements to improve this paper.

2-Title and objective. The term "consumption" might not be relevant for the present study, given that what is measured is more than just "the using up of goods and services". As explained in the introduction, PWS can have difficulties to access to treatment (which is not directly linked to the consumption). Maybe the term "treatment" would be more relevant.

Response: We have changed this term as requested throughout the manuscript.

* Introduction section.

3- P3, l22 "These factors can include dental caries, periodontal diseases" is unnecessary, since explained factors are related to "dental caries and periodontal measurement indexes (l20).

Response: We have deleted this sentence.

4- P3, l28-31. Please indicate in the text that ref 23 is about PWS.

Response: We made this correction in the text of the paper (page 3, line 30).

5- P4, l32-33: instead of "on this subject", indicate "on oral health epidemiology".

Response: This correction has been made (page 4, line 34).

* Material and methods

6- P4, l57: The paragraph should be entitled "Groups constitution", with two clear sub-paragraphs: PWS identification and control identification.

Response: We thank the reviewer for this suggestion and modified this paragraph accordingly. 

7- P5, l70-1: The sentence "Follow-up dental care was explored for all people in the EGB over a period of 3 years (2014 to 2017)" should be placed in the "design of the study and sample of the population" section, because it allows to understand why the study is a retrospective cohort.

Response: We agree with the Reviewer. This correction was made (page 4, lines 43-44).

8-P5, l76: please indicate that these 3 outcomes are covered by French National Insurance

Response: Done (page 5, line 81).

9-P5, l81: "The outcomes of interest were explored as qualitative variable" should rather be ""The outcomes of interest were explored as binary (at least one dental treatment) and ordinal qualitative (1,2-3 or >3) variables." Please make the changes in all the manuscript (e.g. p5, l86, etc)

Response: We thank the Reviewer for this comment. 

We made a mistake in the manuscript: the variables with several categories (1, 2-3 or >3) were considered as qualitative variables, and not as quantitative variables. The outcomes of interest were explored as qualitative nominal variables (binary or > 2 levels). We don’t consider the variables > 2 categories as ordinal qualitative variables. We have clarified this point in the method section. 

Page 5, lines 88-90: “The outcomes of interest were explored as qualitative variables: binary (at least one dental treatment) or nominal variables with more than 2 categories (1, 2-3 or >3).”

10- P5, l87: the Fisher exact test is OK. Pearson χ2 test is not necessary.

Response: We agree, we deleted “Pearson χ2 test” in the sentence.

11- P5, l88: The number and percentage of dental acts were presented ...

We thank the Reviewer for this suggestion. We modified the sentence accordingly: 

Page 6, lines 96-97 “The number and percentage of dental acts were presented by class (1, 2-3, or >3) for each type of care.”

12- P5-6: "First, PWS were localized according to their place of residence. However, mapping by residence may have compromised patient anonymity in a few cases, so we chose to map according to the department of residence." is OK, but should be removed in the text because it doesn't add meaningful information.

Response: This correction was made

13- P6, l96: Paragraph should be entitled "Mapping of population and distribution of dental care" , thus "We mapped the distribution of PWS and the distribution of dental care" l97 should be removed.

Response: This correction was made.

14- P6, l102: As it is an observational study, authors should be cautious with causation. Thus, it should be written something like " To estimate the association between schizophrenia and dental outcomes, etc..."

Response: We thank the Reviewer for this comment. We have made the change as requested (page 6, line 111).

*Results: Results are well reported

15- p7, l137: "confirmed" is not necessary

Response: We deleted “confirmed” as requested

* Discussion: Discussion is well conducted

16 -p8, l180, typo "carries"

Response: Thank you for pointing this error which was corrected.

17- Maybe the paragraph p8, l171-177 should appear in the limitation section of the discussion.

Response: We agree with this suggestion. We moved this sentence from the beginning of the “discussion” section to the” limitation” subsection (page 12, lines 294-300). 

Figures

18-in the legends, please change "à" to "to"

Response: Thank you for pointing this error which was corrected.

19-Table 1: Presentation of p-values should be clearer. Readers should easily understand that there are two types of tests in the table (for binary outcomes and qualitative ordinal outcomes).

As mentioned before, we have considered all the variables as qualitative nominal variables. We have used the Pearson Chi² test or the Fisher exact test, under the conditions of application. The Fischer exact test was added in the table, as you indicated that the Pearson Chi² test was not necessary. 

20-Table 2: Authors should keep the same definitions in the text and in the table. The labeling of "Persons without schizophrenia" is unclear.

Response: We agree with this suggestion. We removed "Persons without schizophrenia” and have replaced it by “Control Group”.

---

## [Editor Report · Decision Letter 1]

19 Feb 2020

Oral health treatment habits of people with schizophrenia in France: a retrospective cohort study

PONE-D-19-20482R1

Dear Dr. Denis,

We are pleased to inform you that your manuscript has been judged scientifically suitable for publication and will be formally accepted for publication once it complies with all outstanding technical requirements.

With kind regards,

Denis Bourgeois

Academic Editor

PLOS ONE
---

## [Editor Report · Acceptance letter]

21 Feb 2020

PONE-D-19-20482R1 

Oral health treatment habits of people with schizophrenia in France: a retrospective cohort study 

Dear Dr. Denis:

I am pleased to inform you that your manuscript has been deemed suitable for publication in PLOS ONE. Congratulations! Your manuscript is now with our production department. 

With kind regards,

on behalf of

Professor Denis Bourgeois 

Academic Editor

PLOS ONE